# What are the social predictors of accident and emergency attendance in disadvantaged neighbourhoods? Results from a cross-sectional household health survey in the north west of England

Clarissa Giebel,[1,2] Jason Cameron McIntyre,[3] Konstantinos Daras,[4] Mark Gabbay,[1,2] Jennifer Downing,[2,5] Munir Pirmohamed,[2,4] Fran Walker,[2] Wojciech Sawicki,[6] Ana Alfirevic,[2,5] Ben Barr[1,2]

For numbered affiliations see end of article.

**Correspondence to**
Dr Clarissa Giebel;
clarissa.giebel@liverpool.ac.uk

## ABSTRACT

**Objectives** The aim of this study was to identify the most important determinants of accident and emergency (A&E) attendance in disadvantaged areas.

**Design, setting and participants** A total of 3510 residents from 20 disadvantaged neighbourhoods in the North West Coast area in England completed a comprehensive public health survey.

**Main outcome measures** Participants were asked to complete general background information, as well as information about their physical health, mental health, lifestyle, social issues, housing and environment, work and finances, and healthcare service usage. Only one resident per household could take part in the survey. Poisson regression analysis was employed to assess the predictors of A&E attendance frequency in the previous 12 months.

**Results** 31.6% of the sample reported having attended A&E in the previous 12 months, ranging from 1 to 95 visits. Controlling for demographic and health factors, not being in employment and living in poor quality housing increased the likelihood of attending an A&E service. Service access was also found to be predictive of A&E attendance insofar as there were an additional 18 fewer A&E attendances per 100 population for each kilometre closer a person lived to a general practitioner (GP) practice, and 3 fewer attendances per 100 population for each kilometre further a person lived from an A&E department.

**Conclusions** This is one of the first surveys to explore a comprehensive set of socio-economic factors as well as proximity to both GP and A&E services as predictors of A&E attendance in disadvantaged areas. Findings from this study suggest the need to address both socioeconomic issues, such as employment and housing quality, as well as structural issues, such as public transport and access to primary care, to reduce the current burden on A&E departments.

## INTRODUCTION

Accident and emergency (A&E) attendance rates are rapidly increasing in the UK and are particularly high in disadvantaged areas.

### Strengths and limitations of this study

► This survey is novel insofar as it examines a comprehensive set of factors linked to accident and emergency attendance, including physical health, mental health, service access and socioeconomic factors.

► Participants were recruited from a wide geographical area of disadvantaged neighbourhoods, which strengthens the representativeness of the survey findings.

► Households were approached at different times during the day to produce a less biased sample.

► Some of the most disadvantaged people may not be captured in the survey due to not having a fixed address.

This is leading to unsustainable pressure on the National Health Service (NHS), with 20.5 million attendances reported in England from April 2015 to March 2016, compared with 19.5 million the previous year.[1] According to a King's Fund report, unnecessary A&E attendances could be reduced by between 8% and 18% if patients attended a primary care service for their health concerns instead.[2] The predictors of A&E attendance are complex and involve multiple components. While we know that A&E attendances are high in disadvantaged areas,[3] the specific reasons for this are less clear. The prevalence of health problems is likely to be important; however, other factors may also play a role. Housing problems and employment status, for example, have been associated with high levels of A&E attendance, as have loneliness and low levels of social support.[4 5] Health service factors, including proximity to A&E and poor access to primary care, have also

BMJ

been linked to higher A&E attendance rates in past research.[3 6–11]

Previous studies on predictors of A&E attendance have been limited because they have not assessed the independent influence of the full range of potential risk factors, including morbidity, socioeconomic factors, social support and proximity to A&E and primary care. In a recent study by Hull and colleagues,[12] primary and secondary care data were linked with A&E attendance data in London. The data showed that multimorbidity, smoking, increased number of general practitioner (GP) attendances, shorter distance to A&E and age were all linked to visits to an A&E department. By comparing the most disadvantaged versus the least disadvantaged quintile, Hull and colleagues[12] reported a 52% higher A&E attendance in the most disadvantaged population sample.

The aim of this study was therefore to use data from a household health survey (HHS) conducted in disadvantaged neighbourhoods in the North West of England to identify the most important determinants of A&E attendance in disadvantaged populations to inform actions that could reduce demand for A&E services.

## METHODS

### Participants

The Northwest Coast area of England contains some of the most disadvantaged neighbourhoods in the country.[13] The HHS of the National Institute for Health Research Collaboration for Leadership in Applied Health Research and Care in the North West Coast (NIHR CLAHRC NWC) was conducted between August 2015 and January 2016 to identify individual and neighbourhood level factors that contribute to inequalities in physical and mental health and to provide a baseline for evaluating neighbourhood level public health interventions.

Twenty disadvantaged neighbourhoods were initially selected by participating local authorities. Each of the 10 local authorities that was part of the NIHR CLAHRC NWC partnership was asked to select a neighbourhood that would be involved in a number of public health initiatives and a comparator area with similar characteristics. The local authorities were asked to select these neighbourhood based on the following criteria: they (1) had a population between 5000 and 10 000 residents; (2) had a deprivation index[13] in the bottom 15% of the national average and (3) had a coherent or shared sense of identity among residents. In addition, the intervention areas needed to have a local community organisation that could support the implementation of public health interventions. The selection of these neighbourhoods was at the discretion of the participating local authorities. The public health initiatives commenced in 10 of the neighbourhoods after the survey data were collected. An additional sample (n=809) was also collected from eight less disadvantaged neighbourhoods, however, data from this sample were not used in this study.

Prior to the survey, ethical approval was obtained from the University of Liverpool (Ref: RETH000836). The survey fieldwork was conducted by BMG research and a pilot survey was completed with 36 residents from the disadvantaged neighbourhoods prior to the full data collection, which resulted in minor changes to survey documents. A random sample of households was then drawn from each of the disadvantaged neighbourhoods and one randomly selected respondent completed a face-to-face interview in each household (n=3510). The survey combined information on a wide range of socioeconomic factors, mental and physical morbidities, and healthcare utilisation. Full details of the survey methodology are published elsewhere.[14]

### Variables

Our outcome variable was defined as the number of times respondents reported attending an A&E department over the previous 12 months. Measures of socioeconomic conditions included education (no qualifications, professional or vocational certificate, degree or higher), employment (working/not working), financial hardship (doing well/getting by, struggling), change in financial circumstances (getting better or unchanged/worse than 12 months ago) and housing quality (no problems/ problems with cold, damp or mould). Physical health was assessed with the four physical health dimensions of the EuroQuol five-dimension scale (EQ-5D)[15] for mobility, self-care, engagement in usual activities and pain (no problems/some or severe problems). Mental health was assessed using a series of validated instruments. Depression was measured using the nine-item Patient Health Questionnaire.[16] Anxiety was measured using the seven-item Generalised Anxiety Disorder scale.[17] Paranoia was measured using the persecution subscale of the persecution and deservedness scale (PaDS) for symptoms of paranoia.[18] Practical social support and social contact were assessed based on the level of agreement with the statements 'If I needed help, there are people who would be there for me' and 'If I wanted company or to socialise, there are people I can call on'. Response options for both items ranged from 1=*definitely disagree* to 4=*definitely agree*. (Further information on each of the survey variables is given in the online supplementary appendix 1).

The proximity to A&E departments and GP practices was estimated using the Routino open source tool 1 (https://www.routino.org/uk/) [1] to calculate the shortest road distance between the centre of each postcode and these health facilities. The average distance for all postcodes within each Lower layer Super Output Area (LSOA) level (LSOA is a geographical area in the UK which is based on postcodes and clusters together groups of the population.)[2] was then estimated and linked to survey responses based on the LSOA in which the respondent lived.

## Public involvement

Members of the public have been involved throughout the design and implementation of the survey, in addition to external partners and representatives from local authorities. This involvement included attending research and design meetings, providing feedback on study documents, as well as helping with the dissemination of findings. Public advisors have been recruited from the sampled disadvantaged areas and receive monetary reimbursement for their involvement.

## Data analysis

Data were analysed using Stata version 14. Because the A&E attendance variable comprised highly skewed ($S$-$W$=0.39, p<0.001) count data, we constructed a Poisson regression model. Standard errors were adjusted to account for the multistage nature of the survey sampling using the *svyset* commands in Stata[19] and the analysis was weighted for non-response. This model provided estimates of the rate ratio (RR) of A&E attendance associated with each variable, while holding all other variables constant in the model. We then used this model to calculate the absolute rate differences (ARD) as the marginal effect of each of the variables while holding the other variables at their means. Demographic, socioeconomic, health condition, disability, symptom and healthcare access variables were entered into the model.

## RESULTS

Of the 3510 survey respondents, 1109 (31.6%) reported attending A&E at least once in the past 12 months. Among those, the number of attendances ranged from 1 to 95 (M=2.70, SD=4.90) and the overall rate of A&E attendance was 75 attendances per 100 population (95% CI 64 to 87).

Table 1 shows the results of the Poisson regression model as well as descriptive statistics for the independent variables. RR and ARD are reported alongside CIs. Age was a significant demographic predictor of A&E attendances. After adjusting for other variables in the model, the A&E attendance rate was three times higher among 18–24 year olds compared with people over 64 years of age (RR=2.72, 95% CI 1.33 to 5.53). In absolute terms, this was equivalent to an additional 60 attendances per 100 18–24 year olds compared with over 65 year olds, other factors being held constant (ARD=60 per 100, 95% CI 10 to 110). The health and disability variables predictive of A&E attendance included having a health condition as opposed to no health condition (RR=1.78, 95% CI 1.19 to 2.66, ARD=28 per 100, 95% CI 10 to 45) and having problems with self-care such as washing and dressing (RR=2.63, 95% CI 1.82 to 3.79; ARD=71 per 100, 95% CI 34 to 109). People with multiple morbidities were not more likely to attend A&E than people with only one condition. Of the socioeconomic and environmental factors significantly associated with A&E attendance, being out of work increased risk of A&E attendance by 38% (RR=1.38, 95% CI 1.08 to 1.78;

ARD=16 per 100, 95% CI 4 to 27) and living in a house with problems with cold, mould or damp increased risk of A&E attendance by 34% (RR=1.34, 95% CI 1.01 to 1.76; ARD=0.14 per 100, 95% CI 1 to 27). Surprisingly in this sample of people living in relatively disadvantaged neighbourhoods, higher levels of education were associated with higher risk of A&E attendance when controlling for other factors. Levels of social support and social contact did not predict A&E attendance.

Both healthcare access variables were significantly related to A&E attendance. Specifically, living further from an A&E department reduced the likelihood of attending an A&E service by 7% per kilometre (RR=0.93, 95% CI 0.89 to 0.97), and living further from a GP increased the likelihood of attending A&E by 46% per kilometre, after controlling for health status, socioeconomic and demographic factors (RR=1.46, 95% CI 1.12 to 1.90). ARDs indicated that for each kilometre closer, the average person lived to an A&E department, they had three fewer A&E attendances per 100 population in the previous 12 months (ARD = −3, 95% CI −5 to −2). Conversely, there were 18 more A&E attendances per 100 population for each kilometre further the average person lived from a GP practice (ARD=18, 95% CI 6 to 30).

## DISCUSSION

This is one of the first surveys to collect data on socioeconomic background, mental and physical health and healthcare utilisation, and to combine these factors with distance to healthcare services. Spanning a wide geographical area in one of the most disadvantaged areas of England,[13] findings from the survey identified several key risk factors of A&E attendance, including young age, depression, high education, non-employment, poor housing, as well as longer distance from a GP and shorter distance to an A&E service.

Considering the increase in A&E attendance across England,[1] particularly in disadvantaged areas,[3] it is important to understand the reasons behind this rise and to design strategies to reduce health inequalities. A&E attendance was significantly higher for 18 to 24 year olds, when controlling for other factors, which has been found in previous work,[20] and may be due to heightened rates of accidents, trauma and alcohol-related attendances at A&E,[21] or higher use of A&E for primary care reasons in this age group. A limitation of our study is that we did not collect any information on the reasons for A&E attendance. Furthermore, A&E attendance has to be distinguished from admission to hospital, which we know is more prevalent in the elderly with long-term conditions.[22]

Higher levels of education were linked to increased risks of attending A&E services in our disadvantaged sample. Considering that people from a low socioeconomic background are more likely to attend A&E services, as shown in this survey and supported by other research,[23–27] it is perhaps surprising to find those with a professional or degree level education to be up to twice as likely to access

**Table 1** Determinants of number of A&E attendances over the previous 12 months (n=3510)

| Socioeconomic factors | Adjusted rate ratio of A&E attendance | Adjusted rate ratio 95% CI | Absolute risk difference of A&E attendance | Absolute risk difference 95% CI | Unadjusted rate ratios (95% CI) | N (%) | M (SD) |
|---|---|---|---|---|---|---|---|
| Demographics | | | | | | | |
| Age (65+) (years) | – | – | – | – | – | 858 (24.5) | – |
| 18–24 | 2.72** | 1.33 to 5.53 | 0.60* | 0.10 to 1.10 | 1.15 (0.70 to 1.88) | 368 (10.5) | – |
| 25–44 | 1.43 | 0.95 to 2.16 | 0.15 | –0.01 to 0.31 | 0.83 (0.62 to 1.12) | 1226 (34.9) | – |
| 45–64 | 1.21 | 0.83 to 1.77 | 0.07 | –0.07 to 0.21 | 1.18 (0.81 to 1.72) | 1057 (30.1) | – |
| Gender (female) | 0.9 | 0.69 to 1.18 | –0.05 | –0.18 to 0.08 | 0.89 (0.66 to 1.20) | 2029 (57.8) | – |
| Ethnicity (BME) | 0.6 | 0.35 to 1.04 | –0.20* | –0.37 to –0.03 | 0.40 (0.24 to.67) | 366 (10.5) | – |
| Socioeconomic status | | | | | | | |
| Education (no qualifications) | | | | | | 1516 (43.3) | – |
| Professional/vocational certificate | 1.58** | 1.17 to 2.14 | 0.20** | 0.07 to 0.33 | 1.20 (0.88 to 1.63) | 1579 (45.1) | – |
| Degree or higher | 2.00* | 1.15 to 3.48 | 0.35* | 0.02 to 0.67 | 1.21 (0.71 to 2.07) | 405 (11.6) | – |
| Non-employment (not in paid/self-employment) | 1.38* | 1.08 to 1.78 | 0.16** | 0.04 to 0.27 | 2.20 (1.65 to 2.93) | 2150 (61.3) | – |
| Index of multiple deprivation | 0.99 | 0.99 to 1.00 | –0.003 | –0.01 to 0.001 | 0.99 (0.99 to 1.01) | – | 47.20 (16.90) |
| Financial struggle ('doing well') | | | | | | 716 (20.4) | – |
| 'Getting by' | 1 | 0.72 to 1.40 | <0.001 | –0.16 to 0.16 | 1.33 (0.96 to 1.86) | 2307 (65.8) | – |
| 'Struggling' | 1.22 | 0.75 to 1.98 | 0.1 | –0.16 to 0.36 | 2.94 (1.78 to 4.85) | 481 (13.7) | – |
| Financial situation worse than 12 months ago | 0.99 | 0.78 to 1.26 | –0.004 | –0.12 to 0.11 | 1.49 (1.07 to 2.09) | 578 (16.6) | – |
| Housing quality | | | | | | | |
| Problems with condensation/mould/temperature | 1.34* | 1.01 to 1.76 | 0.14* | 0.01 to 0.27 | 1.73 (1.26 to 2.38) | 1138 (33.7) | – |
| Health status | | | | | | | |
| Mobility problems (EQ5D) | 1.24 | 0.84 to 1.83 | 0.11 | –0.10 to 0.32 | 3.54 (2.64 to 4.76) | 930 (73.5) | – |
| Self-care problems (EQ5D) | 2.63*** | 1.82 to 3.79 | 0.71*** | 0.34, 1.09 | 5.58 (3.85 to 8.09) | 381 (10.9) | – |
| Problems engaging in usual activities (EQ5D) | 1.43* | 1.01 to 2.03 | 0.19 | –0.02 to 0.40 | 4.17 (3.12 to 5.57) | 852 (24.3) | – |
| Pain (EQ5D) | 1.24 | 0.90 to 1.71 | 0.11 | –0.05 to 0.27 | 3.30 (2.50 to 4.36) | 1325 (37.8) | – |
| 1 condition (base category: no condition) | 1.78** | 1.19 to 2.66 | 0.28** | 0.10 to 0.45 | 3.19 (2.38 to 4.27) | 857 (24.4) | – |
| >1 Condition (base category: 1 condition) | 1.02 | 0.67 to 1.55 | 0.01 | –0.19 to 0.21 | 2.80 (2.08 to 3.77) | 1256 (35.8) | – |
| Mental health | | | | | | | |
| Depression (PHQ–9) | 1.36* | 1.01 to 1.83 | 0.15 | –0.003 to 0.30 | 2.02 (1.72 to 2.37) | – | 1.56 (0.68) |
| Anxiety (GAD-7) | 0.86 | 0.65 to 1.14 | –0.07 | –0.21 to 0.06 | 1.75 (1.47 to 2.07) | – | 1.54 (0.75) |
| Paranoia (PaDS–5) | 1.04 | 0.89 to 1.21 | 0.02 | –0.06 to 0.09 | 1.41 (1.21 to 1.65) | – | 2.01 (0.89) |

Continued

**Table 1** Continued

| Socioeconomic factors | Adjusted rate ratio of A&E attendance | Adjusted rate ratio 95% CI | Absolute risk difference of A&E attendance | Absolute risk difference 95% CI | Unadjusted rate ratios (95% CI) | N (%) | M (SD) |
|---|---|---|---|---|---|---|---|
| **Social support** | | | | | | | |
| Practical support | 0.93 | 0.47 to 1.85 | −0.03 | −0.36 to 0. 29 | 0.69 (0.30 to 1.59) | 3302 (94.3) | – |
| Social contact | 1.08 | 0.70 to 1.70 | 0.04 | −0.17 to 0. 25 | 0.64 (0.31 to 1.31) | 3286 (94.0) | – |
| **Healthcare access** | | | | | | | |
| Distance to GP (km) | 1.46** | 1.12 to 1.90 | 0.18** | 0.06 to 0. 30 | 1.21 (0.96 to 1.53) | | 0.99 (0.66) |
| Distance to A&E (km) | 0.93*** | 0.89 to 0. 97 | −0.03*** | −0.05 to −0.02 | 0.95 (0.91 to 0.99) | | 7.12 (4.21) |

Risk ratios and absolute risk differences are adjusted for all other variables in the model.
*p<0.05, ** p<0.01, *** p<0.001.
BME, Black and Minority Ethnic; EQ5D, Euroqol five dimensions; GAD-7, Generalised Anxiety Disorder Assessment; PaDS-5, Persecution and Deservedness Scale; PHQ-9, Patient Health Questionnaire.

A&E when holding all other factors constant. It should be noted however that this is only the case when adjusting for other health and socioeconomic factors—the crude risk ratios for education were all not significantly different from 1. So it appears in our sample that where people with higher levels of education are living in disadvantaged areas and have similar levels of health problems as the rest of the population and are living in similar social circumstances, they are more likely to attend A&E than there less educated peers. One possible reason for this may be that higher educated residents are more likely to recognise specific symptoms and become proactive by seeking medical help. It is also plausible that people with higher qualifications have ended up living in the disadvantaged neighbourhood, due to their poor health and this leads to increased A&E attendances. Indeed, depression and problems with self-care were independently associated with more A&E visits, supporting previous evidence of increased emergency attendances among people with mental health problems who are more likely to be 'frequent users'.[6 28 29] Thus, interventions that address mental health problems, particularly depression, and that provide greater support to people who have problems with self-care may be linked to concomitant declines in A&E attendance. However, the relationship between depression and A&E attendance should be interpreted with some caution because the RR was found to be significant, while the absolute risk difference was not significant.

Being out of work and living in poor housing were important predictors of increased A&E attendance, above and beyond the effect of health status. This could be because these factors exacerbate existing conditions. Indeed, unemployment has been linked to higher A&E attendances for asthma,[30] while poor housing quality has been linked to more asthma-related symptoms[31] and worse mental health.[32] Controlling for these socioeconomic factors, the Index of Multiple Deprivation (IMD) was not independently associated with A&E attendance within this sample of disadvantaged areas. (To check for multicollinearity we regressed each individual socioeconomic variable onto IMD, using dummy coding for categorical predictors. All standardised coefficients were <0.26, suggesting that IMD was not collinear with the other socioeconomic measures.)[3] This indicates that much of the high level of A&E attendance found by others in disadvantaged areas[3] is possibly explained by differences in the prevalence of health conditions, high unemployment and poor housing. This lack of an effect related to deprivation may also be related to the limited variation in deprivation between the relatively deprived areas in our sample. In contrast, social support was not found to be linked to A&E attendance. While a strong social network offering practical and emotional support is important for well-being,[33] this did not affect the attendance of A&E services in our study. It may be the case that participants may have genuinely needed to access healthcare services for their physical and mental health needs which could

not be supported otherwise. Therefore, interventions that improve housing conditions and promote employment in disadvantaged communities should be integral to strategies to reduce demand for A&E services.

We found that a longer distance from GP practices and a shorter distance to A&E departments were significant predictors of A&E attendance. The findings are consistent with previous work on A&E proximity[8 9 12]; however, we additionally identified GP practice proximity as a unique determinant of A&E attendance. The results indicate that primary care access predicts A&E attendance over and above the effects of health status, socioeconomic status and A&E access. The current trend of consolidating GP practices in fewer, larger health centres may therefore be contributing to increases in A&E attendances. This trend may also be linked to ease of access, in that patients may be unable to get a GP appointment and therefore attend an A&E service instead. Future approaches should be cognisant of primary care service placement and access. Reduced demand for A&E may be achieved via improved public transport links that consider primary care locations or via strategically positioning primary care services in areas with high A&E attendance rates. It would be of interest in future research to examine whether factors such as transport links or appointment availability independently predict A&E attendance or, in the case of public transport links, explain the relationship between distance to services and service usage.

One recently conducted retrospective study has also explored the socioeconomic predictors of A&E attendance,[12] showing that A&E attendance increased with increasing numbers of health conditions, in addition to the risk of A&E attendance increase with smoking, GP attendance, being housebound and age. However, the study differed in several aspects from the HHS. On the one hand, the study by Hull and colleagues[12] did not measure the effects of proximity to GP services. In addition, our HHS contains a rich set of mental health, health-related quality of life and socioeconomic variables, whereas the London-based assessment of primary and secondary care data only employs the IMD as a measure of social deprivation, having no information on, for example, housing quality, while health measures were limited to numbers of conditions. In our study, A&E attendance did not increase with increasing numbers of health conditions. One plausible explanation for this effect is that the number of health conditions a person has is not in itself a risk for A&E attendance, but this is just a marker of people with poorer mental health and health-related quality of life. Further research is required to confirm whether this explains the divergent findings.

The findings from this survey should be considered in light of a few limitations. The cross-sectional nature of the survey only allowed investigation of the predictors of A&E attendance at one point in time. Moreover, this survey was based on self-reports, and thus participants had to remember whether they had attended an A&E department, and how often, in the past 12 months.

For this reason, there could potentially have been some recall bias in the number of A&E attendances. This could be avoided by referring to Hospital Episodes Statistics data from the National Health Service. The survey also purposefully recruited residents from some of the most disadvantaged areas within England and the Northwest Coast and selected only neighbourhoods that met criteria that included methodological (eg, large population and shared neighbourhood identity) and practical (eg, infrastructure for interventions) constraints. Thus, the findings may not generalise to other areas that did not meet these criteria. Examining such a wide range of determinants meant striking a balance between including multiple determinants and examining those determinants in detail. Because of this, examining more nuanced research questions, such as the role of different types of health conditions on A&E attendance, was not possible. Lastly, people without a fixed address who may be living in extremely disadvantaged circumstances were probably not captured by our data.

## CONCLUSIONS

We conducted a cross-sectional analysis of A&E attendance risk factors in a large sample of UK residents in disadvantaged neighbourhoods. To our knowledge, this is the most comprehensive examination of diverse socioeconomic determinants of A&E attendance, and the first to examine GP proximity alongside A&E proximity. Housing quality, employment status, self-care difficulties, depression and proximity to both primary and emergency services were identified as the most important determinants of A&E attendance. The findings highlight the need for multipronged approaches to reduce A&E attendances that address socioeconomic inequalities, such as employment and housing quality, but also structural issues such as access to primary care. Key policy suggestions likely to reduce A&E attendance based on these findings include: strategic placement of—and better transport links to—primary care services, better housing conditions, increased employment and improved support for self-care and public mental health. Improving housing conditions has shown short-term benefits on the health of residents, without having evaluated A&E attendance rates.[31 34] To date, findings on the effect of improving primary care access interventions on reducing A&E attendance appear inconclusive,[35] suggesting the need for further implementation strategies to improve primary care access. Strategies to reduce strain on health services need to address the social and economic factors that underpin demand for healthcare and enhance access to primary care.

**Author affiliations**
[1]Institute of Psychology, Health and Society, University of Liverpool, Liverpool, UK
[2]NIHR Collaboration for Leadership in Applied Health Research and Care, North West Coast, UK
[3]School of Natural Sciences and Psychology, Liverpool John Moore's University, Liverpool, UK

[4]Geographic Data Science Lab, University of Liverpool, Liverpool, UK
[5]Institute of Translational Medicine, The University of Liverpool, Liverpool, UK
[6]Royal Liverpool and Broadgreen University Hospitals NHS Trust, Liverpool, UK

**Contributors** CG and JMI jointly wrote the article. JMI and BB performed statistical analysis. KD, MP, MG, JD, FW, WS, AA and BB edited and provided feedback on drafts of the manuscript and approved the final version.

**Funding** This work was supported by the National Institute for Health Research, Collaboration for Leadership and Health Research and Care North West Coast (NIHR CLAHRC NWC) and the state Department of Health and Social Care.

**Disclaimer** The views expressed are those of the author(s) and not necessarily those of the NHS, the NIHR or the Department of Health.

**Competing interests** We have read and understood BMJ policy on declaration of interests and declare the following: All authors have completed the ICMJE uniform disclosure form.

**Patient consent for publication** Not required.

**Ethics approval** The research was approved by the University of Liverpool Committee on Research Ethics (Ref: RETH00836).

**Provenance and peer review** Not commissioned; externally peer reviewed.

**Data sharing statement** The dataset from the North West Coast Household Health Survey will be made publicly available after an embargo period.

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
