## [Reviewer comments · BMJ Open]

This paper was submitted to a another journal from BMJ but declined for publication following peer review. The authors addressed the reviewers' comments and submitted the revised paper to BMJ Open. The paper was subsequently accepted for publication at BMJ Open.

(This paper received three reviews from its previous journal but only two reviewers agreed to published their review.)

ARTICLE DETAILS

TITLE (PROVISIONAL)	What are the social predictors of Accident & Emergency attendance in disadvantaged neighbourhoods? Results from a cross-sectional household health survey in the North West of England
AUTHORS	Giebel, Clarissa; McIntyre, Jason Cameron; Daras, Konstantinos; Gabbay, Mark; Downing, Jennifer; Pirmohamed, Munir; Walker, Fran; Sawicki, Wojciech; Alfirevic, Ana; Barr, Ben

VERSION 1 – REVIEW

REVIEWER	Jessica Sheringham & Helen McDonald UCL, UK
REVIEW RETURNED	05-Apr-2018

GENERAL COMMENTS	Thank you for inviting me to review this paper. I have conducted the review in collaboration with my colleague, Dr Helen McDonald, so the text below summarises both our perspectives on the paper. As required by the review criteria the comments relate to scientific credibility and research/publication ethics rather than judgement on the significance of the study. This study describes the association of a wide range of factors (including physical health, mental health, health service proximity, housing quality, finances, employment status and social support) with A&E attendance in the previous 12 months. It is based on a cross-sectional survey of households in deprived areas. A key strength of the study is the wide range of factors that are considered together, including several that are difficult to obtain from routinely collected data such as self-assessed degree of financial struggle, housing quality, and pain. The manuscript is easy to read and the results are presented clearly. Our major comment about the entire paper centres on what can be generalised from this particular study population and several of our comments below relate to this. Major points AIM/PURPOSE  From the introduction, we understand this study is seeking to explain why A&E attendances are high in deprived areas (compared with non-deprived areas?). The sample, however only comes from deprived areas, with particular characteristics – i.e. in the most deprived IMD
---

- decile, strong social cohesion and with resilience infrastructure. These particular characteristics mean:
- a. it is difficult to generalise *beyond* this population to explain why A&E attendance may be higher in deprived areas.
 - b. it might indicate why certain of the associations are counter to what has been found in other studies (eg concerning associations with social support/contact)
 - c. it might be unproductive to examine associations with some variables where the range is small, e.g. this applies to IMD, the range of IMD scores determined by the sample is limited to one decile only. It might also apply to the other socioeconomic variables e.g. education.

The study does still offer opportunities to understand *within* a deprived population what factors are associated with higher self-reported A&E use. Given the comparatively high A&E attendance amongst individuals living in deprived areas, addressing this question would still provide a valuable contribution to the field.

METHODS

2. We appreciate details of the methods are published elsewhere but further information on the sample would enable readers to judge the scientific credibility of the study. In particular, it would be useful to have information on:
 - a. Selection of neighbourhoods: Could the authors outline the rationale for the selection criteria, and how was criterion 4 (“had a coherent or shared sense of identity”) defined and identified? How many neighbourhoods were candidates for selection?
 - b. Selection of participants: If the number of participants in this study (N=3,510) differs from that in reference 14, could the authors note if there were any additional exclusion criteria such as missing data, or provide a flow diagram of selection?
 - c. Measures used to obtain information on key variables: for social contact and social support, were validated measures used to assess them?
3. Analysis:
 - a. Methods page 7 line 5 “... variables were entered into the model as predictors of number of A&E attendances over the previous 12 months”. In some places the text (and Table 1) seems to report on a binary outcome of whether the participant attended A&E at least once in the previous 12 months. Was there a separate model with an outcome of number of A&E attendances, and how was this modelled?

RESULTS

4. Could the authors provide a baseline description of the population? For example, an additional column in table 1 with a summary measure for each risk factor (e.g. the number of participants reporting each risk factor, mean distance in km to healthcare)? The prevalence of the risk factors would be helpful in assessing:
 - a. how likely the results are to be generalisable to other populations, and

- b. which factors might contribute most to A&E attendance.

DISCUSSION

5. Do the authors consider the selection criteria may affect generalisability across deprived areas? For example, might individual level social support be more relevant to A&E attendance in neighbourhoods not selected for a coherent or shared sense of identity?
6. The authors have acknowledged a key limitation, ie they don't have the reason for the A&E attendance or subsequent service use (eg if attendance resulted in an admission). This is important because it tells about the extent to which these attendances could be preventable. They make a reasonable inference that in younger populations attendances might due to accidents, trauma, and alcohol-related incidents. It might be expected therefore that associations are different in a younger population where attendances were primarily for acute reasons vs an older population where more attendances are driven by exacerbations of chronic health conditions. If the sample size allows, the authors could explore this by re-analysing the data stratified by age (possibly using cut-offs informed by national data on the reason for A&E attendances).

Minor points

There were some minor points where clarification may be helpful:

1. Abstract: it would be helpful to give some numbers in the results section for the reader to judge the size of variation in attendances between groups with different characteristics.
2. Methods page 6 line 52 "Standard errors were adjusted to account for the multi-stage nature of the survey sampling" How were standard errors adjusted, and could the authors briefly describe the multi-stage nature of the sampling?
3. Did the authors consider taking clustering by LSOA into account for analyses which included distance to healthcare services, or did the sampling prevent there being multiple participants within each LSOA?
4. Results page 7 line 14: "living further from a GP ($p < 0.05$) increased the likelihood of attending A&E by 28 percent per kilometer, after controlling for health status, socioeconomic and demographic factors". Please could the authors clarify how this relates to the odds ratio of 1.46 in the penultimate row of table 1?
5. Discussion page 8 lines 16 and 26 "more A&E visits" and "increased A&E attendance". If the outcome was a binary outcome of at least one A&E attendance, rather than the number of A&E visits, it might be clearer to write "A&E visits".
6. Discussion page 9 line 35-48. The authors suggest that the lack of association of IMD with A&E attendance within this sample of highly deprived areas "indicates that much of the high level of A&E attendance found by others in deprived areas is probably explained by differences in the prevalence of health conditions, high unemployment and poor housing". All the neighbourhoods included in this study were in the bottom 10% of the national IMD, which might reduce any observed association of IMD with outcomes in this study, and so this interpretation seems a little strong. If the authors are interested in how much of the A&E attendance is explained by the factors

	in the model, have they considered other approaches to look at this such as goodness of fit? 7. Discussion page 10 lines 20-35. It might be useful for the general reader to include a comparison of this study's findings with the findings in the study by Hull et al. 8. Conclusions page 10 lines 18-28. "The findings highlight the need... and public mental health." The authors call for strategies to reduce A&E attendances based on these factors. While they have demonstrated strong associations (consistent with previous evidence that did not consider all these factors together) and discussed plausible causal pathways for these, it might strengthen their conclusions to add reference to evidence that these factors can be successfully addressed with interventions. 9. Table 1: are all estimates are mutually adjusted? Are the odds ratios for healthcare access are per kilometre? As above, it would be useful to add baseline population characteristics and unadjusted estimates to this table too.
--	--

REVIEWER	Bernard Silke St James' Hospital and Trinity College, Dublin
REVIEW RETURNED	18-Apr-2018

GENERAL COMMENTS	This study considers the social predictors of A&E attendance in deprived neighbourhoods using a cross-sectional household health survey in the North West of England. Of course the self-reported data may not be identical to those data captured by a national census; nonetheless the approach was structured, with an appropriate sample and adequate detail. There were few surprises. Multi-morbidity in younger persons from areas of low SES with depression has been described; the authors found that being younger and having depression were both associated with higher levels of Accident and Emergency (A&E) attendance among residents from deprived neighbourhood. However, what was surprising is that attend A&E services, as shown in this survey and supported by other research [18], it is perhaps surprising to find those with a professional or degree level education to be up to twice as likely to access A&E. This would be completely the opposite that we have found. Of course the methodology using the Multiple Deprivation Indices may not have the same classification as the classical methods such as Townsend or Carstairs and it is possible that this finding is not totally reliable. Certainly the authors should reference other work suggesting that a higher education leads to a reduction in ED utilisation. Of course many of the findings regarding Deprivation and difficult to solve; public infrastructure such as public transport links or distance to practices are deeply embedded within the structure of society. It is not possible to rectify these perceived deficits readily - not alone due to financial constraints but because the kindness of strangers may lead to a culture of dependence and not necessarily achieve the desired outcomes. Nonetheless this paper is well written can be recommended.
---

REVIEWER	Sally Hull Centre for Primary Care and Public Health Queen Mary, University of London UK I have published several research articles on AED attendance.
REVIEW RETURNED	29-Apr-2018

GENERAL COMMENTS	3. Is the study design appropriate to answer the research question? AED attendance is a topic of general interest. The project uses interview data (collected for another purpose) from individuals in areas with IMD scores in the top decile of deprivation, with the stated aim of identifying the most important determinants of A&E attendance in deprived neighbourhoods. The original survey neighbourhoods (deprived areas defined by IMD centile) were selected for their likelihood of response to a resilience intervention and where there was already a shared sense of identity. Hence this was not a random selection of households, nor a whole locality analysis. The selection process of locations may potentially be a source of bias. There is little detail provided on the survey questions asked, and the signposting to ref. 14 for further details does not elaborate on this. There is particularly little detail on the health questions asked. Which health conditions were included? were they long term conditions? If so how were they categorised? This is important as much of the AED medical literature uses the Quality and Outcomes long term conditions domains, which include both depression and severe mental illness. Is a single category of BME (for ethnicity) useful in view of the varied impact of ethnicity on A&E attendance found in other studies? In measuring distance to the AED – was there only one unit? Or was there a choice of several hospitals for people to attend? How was this handled in the analysis? Why was IMD included in the regression analysis when participants were chosen for this study on the basis that they lived in localities in the top decile of IMD scores? 4. Are the methods described sufficiently to allow the study to be repeated? Better access to the survey questions is needed. There was no independent verification of the outcome measure (numbers of A&E attendances in the previous year) this was based on individual recall alone. 7. If statistics are used are they appropriate and described fully? It would be helpful to see an initial table setting out the informant characteristics from which the logistic regression was derived. It is always useful to see how many individuals there are in the different age groups, and in the other categories.
--

	In the text it states that living further from the GP surgery increased AED attendance by 28% per KM, in the table it states 46%. 10. Are the results presented clearly? As indicated above, it is important to have a table of informant characteristics. 11. Are the discussion and conclusions justified by the results While it is very useful to have individual level data on socioeconomic factors (compared to other studies which rely on aggregate IMD based on postcode/LSOA) the value of this was somewhat masked by the lack of correspondingly detailed information on morbidity, AED attendance and a lack of primary care attendance data. The findings suggest that poorer housing and unemployment are the major factors that explain the IMD element of increased attendance in other studies. These factors are already captured within the IMD score. Proximity to the AED unit is well known as a factor which increases attendance. It was interesting to see that increasing distance from the GP practice was associated with attendance for this patient group. However access to primary care includes appointment availability as well as geographic distance, it would have been useful to link these data to the GP patient survey which provides practice level data on these factors.
--	---

VERSION 1 – AUTHOR RESPONSE

Reviewer 1: We appreciate details of the methods are published elsewhere but further information on the sample would enable readers to judge the scientific credibility of the study. In particular, it would be useful to have information on: a. Selection of neighbourhoods: Could the authors outline the rationale for the selection criteria, and how was criterion 4 (“had a coherent or shared sense of identity”) defined and identified? How many neighbourhoods were candidates for selection? b. Selection of participants: If the number of participants in this study (N=3,510) differs from that in reference 14, could the authors note if there were any additional exclusion criteria such as missing data, or provide a flow diagram of selection? c. Measures used to obtain information on key variables: for social contact and social support, were validated measures used to assess them?	a) We have added and redrafted the section outlining the survey methodology to clarify how the neighbourhoods were selected. The neighbourhoods were selected to provide baseline data for a separate public health intervention which commenced after the survey data was collected. The selection of the neighbourhoods was left to participating local authorities and relied on their local knowledge. We have now added this information to the method section on pages 4 and 5. b) We have outlined in the results that an additional sample (N=809) was obtained from 8 more affluent neighbourhoods as part of the CLAHRC NWC HHS; however, as our focus was on disadvantaged neighbourhoods these were not included in this analysis, so the total including this sample is 3,510+ 809=4319 as reported in reference 14. c) All variables reported in this study have been validated in peer-reviewed journals and/or are used in major national surveys. We have added details on the variables to the methods section and added an appendix with further details on the survey variables.
Methods page 7 line 5 “... variables were entered into the model as predictors of number of A&E attendances over the previous 12	The variable we used for A&E attendance was not binary but continuous. We have clarified this on page 7. As stated in the legend of Figure 1,

months". In some places the text (and Table 1) seems to report on a binary outcome of whether the participant attended A&E at least once in the previous 12 months. Was there a separate model with an outcome of number of A&E attendances, and how was this modelled?	the statistics refer to the number of A&E attendances. We have looked through the text again, and to ensure that this is clear, we have clarified this under 'data analysis'.
Could the authors provide a baseline description of the population? For example, an additional column in table 1 with a summary measure for each risk factor (e.g. the number of participants reporting each risk factor, mean distance in km to healthcare)? The prevalence of the risk factors would be helpful in assessing: a. how likely the results are to be generalisable to other populations, and b. which factors might contribute most to A&E attendance.	We agree that this would help readers interpret the findings and thank Reviewer 1 for the suggestion. The relevant summary statistics (number/proportion or mean/SD) are now reported in Table 1.
Do the authors consider the selection criteria may affect generalisability across deprived areas? For example, might individual level social support be more relevant to A&E attendance in neighbourhoods not selected for a coherent or shared sense of identity?	This is a valid point and indeed the findings may not generalise to all disadvantaged neighbourhoods and we have added discussion of this as a limitation of the present study on page 10.
The authors have acknowledged a key limitation, ie they don't have the reason for the A&E attendance or subsequent service use (eg if attendance resulted in an admission). This is important because it tells about the extent to which these attendances could be preventable. They make a reasonable inference that in younger populations attendances might due to accidents, trauma, and alcohol-related incidents. It might be expected therefore that associations are different in a younger population where attendances were primarily for acute reasons vs an older population where more attendances are driven by exacerbations of chronic health conditions. If the sample size allows, the authors could explore this by re-analysing the data stratified by age (possibly using cut-offs informed by national data on the reason for A&E attendances.	We agree that the patterns of findings may differ across demographic variables - age in particular - and that readers may be interested in these analyses. Analysing interactions between age and other variables would be one way to investigate this. However, as we did not have any a priori hypothesis concerning the interaction between age and any of our exposure variables, testing multiple interactions would be likely to produce spurious significant results, and therefore we haven't included interaction analysis in this study.
Abstract: it would be helpful to give some numbers in the results section for the reader to judge the size of variation in attendances between groups with different characteristics.	We have now stated this information in the abstract.
Methods page 6 line 52 "Standard errors were adjusted to account for the multi-stage nature of the survey sampling" How were standard errors adjusted, and could the authors briefly describe the multi-stage nature of the sampling?	The standard errors are adjusted to take into account that the variance may be due to differences between neighbourhoods compared to within neighbourhoods. i.e they are robust to this source of heteroscedasticity. We do this using the svy commands in Stata. By default, svy computes standard errors by using the linearized variance estimator, otherwise known as a robust variance estimator or Huber/White/sandwich estimator, which is consistent under heteroscedasticity. We have added to the text expanding on this on pages 7.

Did the authors consider taking clustering by LSOA into account for analyses which included distance to healthcare services, or did the sampling prevent there being multiple participants within each LSOA?	We have taken into account the clustering of the data within neighbourhoods as outlined above. The Neighbourhoods contain, on average, 5 LSOAs. There is no reason to think that there would have been any clustering in the variance between the contiguous LSOAs within neighbourhoods, therefore the clustering at the neighbourhood level is sufficient to address this issue.
Results page 7 line 14: “living further from a GP ($p < 0.05$) increased the likelihood of attending A&E by 28 percent per kilometre, after controlling for health status, socioeconomic and demographic factors”. Please could the authors clarify how this relates to the odds ratio of 1.46 in the penultimate row of table 1?	We thank Reviewer 1 for picking up this typo. We have adjusted the numbers on page 9 to accurately reflect the analysis and table.
Discussion page 8 lines 16 and 26 “more A&E visits” and “increased A&E attendance”. If the outcome was a binary outcome of at least one A&E attendance, rather than the number of A&E visits, it might be clearer to write “A&E visits”.	We have clarified that the outcome was the number of A&E visits, not at least one A&E visit.
Discussion page 9 line 35-48. The authors suggest that the lack of association of IMD with A&E attendance within this sample of highly deprived areas “indicates that much of the high level of A&E attendance found by others in deprived areas is probably explained by differences in the prevalence of health conditions, high unemployment and poor housing”. All the neighbourhoods included in this study were in the bottom 10% of the national IMD, which might reduce any observed association of IMD with outcomes in this study, and so this interpretation seems a little strong. If the authors are interested in how much of the A&E attendance is explained by the factors in the model, have they considered other approaches to look at this such as goodness of fit?	We agree that one reason there is not an association with deprivation in our model could be because we are only including more deprived areas (in the bottom 15%). There is however still quite a lot of variation in IMD within the sample ($M = 47.20$, $SD = 16.90$) and our results suggest that within this relatively deprived population it was not an important predictor when controlling for other individual measures of deprivation and housing problems. We have added to the interpretation of this point to reflect the reviewers point (page 11). In relation to the second point, we have not presented measures of model fit as we were not comparing alternative models; rather, we aimed to identify the effects of individual predictors.
Discussion page 10 lines 20-35. It might be useful for the general reader to include a comparison of this study’s findings with the findings in the study by Hull et al.	We have now included more detail on the predictive factors of A&E attendance reported in the Hull et al. (2018) study, whilst leaving in the comparisons between our study and the Hull et al. study on how both differ from one another.
Conclusions page 10 lines 18-28. “The findings highlight the need... and public mental health.” The authors call for strategies to reduce A&E attendances based on these factors. While they have demonstrated strong associations (consistent with previous evidence that did not consider all these factors together) and discussed plausible causal pathways for these, it might strengthen their conclusions to add reference to evidence that these factors can be successfully addressed with interventions.	We have now added information in the conclusions indicating the evidence-base on the efficacy of some of the suggested interventions to reduce A&E attendance rates.
Table 1: are all estimates are mutually adjusted? Are the odds ratios for healthcare access are per kilometre? As above, it would be useful to add baseline population characteristics and unadjusted estimates to this table too.	All estimates in Table 1 are adjusted for all other variables in the model. We have now also included a column of unadjusted coefficients and made it clear (in the results and the table

	heading) which figures are adjusted and which are unadjusted.
Reviewer 2:	
However, what was surprising is that attend A&E services, as shown in this survey and supported by other research [18], it is perhaps surprising to find those with a professional or degree level education to be up to twice as likely to access A&E. This would be completely the opposite that we have found. Of course the methodology using the Multiple Deprivation Indices may not have the same classification as the classical methods such as Townsend or Carstairs and it is possible that this finding is not totally reliable. Certainly the authors should reference other work suggesting that a higher education leads to a reduction in ED utilisation.	We agree that this is a surprising finding, which has not previously been found in the literature. There are a number of potential reasons for this. Firstly, as our sample comprised people from relatively deprived areas, the effect of education may be different in this population compared to the general population. Secondly, we also controlled for a larger number of other measures of socioeconomic and health conditions compared with previous studies. It should be noted that the unadjusted risk ratios indicate that there is no association between education and A&E attendance – but when holding other factors constant, the higher educated participants tended to have a higher rate of attendance. It may be that once you adjust for the higher employment, lower morbidity and better housing conditions, more educated groups are more likely to attend A&E. We have added to the discussion on this point (page 11).
Reviewer 3:	
There is little detail provided on the survey questions asked, and the signposting to ref. 14 for further details does not elaborate on this. There is particularly little detail on the health questions asked. Which health conditions were included? were they long term conditions? If so how were they categorised? This is important as much of the A&E medical literature uses the Quality and Outcomes long term conditions domains, which include both depression and severe mental illness. Is a single category of BME (for ethnicity) useful in view of the varied impact of ethnicity on A&E attendance found in other studies?	We have added further detail on the survey question to the methods on page 6 and have added a web appendix giving more detailed information. The ethnicity variable was categorised into two groups because only 10% of the sample were from a BME population (reflecting populations of these neighbourhoods). This provided insufficient sample to investigate differences between BME groups.
In measuring distance to the A&E – was there only one unit? Or was there a choice of several hospitals for people to attend? How was this handled in the analysis?	We did not take into account the choice of several hospitals for people to attend, considering the fact that when people are in an emergency, they are most likely to be looking for the nearest A&E department.
Why was IMD included in the regression analysis when participants were chosen for this study on the basis that they lived in localities in the top decile of IMD scores?	IMD was included as there was variation in IMD between neighbourhoods in the sample. Further, given that other measures of disadvantage were taken at the individual level, we wanted to control for area level effects that may have additional explanatory power on A&E attendances.
Better access to the survey questions is needed. There was no independent verification of the outcome measure (numbers of A&E attendances in the previous year) this was based on individual recall alone.	We have highlighted the limitations of self-reported attendance. We have also now provided an appendix providing more detail about the survey questions.

It would be helpful to see an initial table setting out the informant characteristics from which the logistic regression was derived. It is always useful to see how many individuals there are in the different age groups, and in the other categories. In the text it states that living further from the GP surgery increased A&E attendance by 28% per KM, in the table it states 46%.	See comment from Reviewer 1 addressing these points.
While it is very useful to have individual level data on socioeconomic factors (compared to other studies which rely on aggregate IMD based on postcode/LSOA) the value of this was somewhat masked by the lack of correspondingly detailed information on morbidity, A&E attendance and a lack of primary care attendance data. The findings suggest that poorer housing and unemployment are the major factors that explain the IMD element of increased attendance in other studies. These factors are already captured within the IMD score. Proximity to the A&E unit is well known as a factor which increases attendance. It was interesting to see that increasing distance from the GP practice was associated with attendance for this patient group. However access to primary care includes appointment availability as well as geographic distance, it would have been useful to link these data to the GP patient survey which provides practice level data on these factors.	We agree that obtaining individual level socioeconomic data was a strength of our study. Other studies only using area based measures will be limited by ecological fallacies. We also have relatively detailed information on health status, including multiple measures of health related quality of life, three mental health instruments and the numbers of health conditions reported. We did not, however, have data on the particular reasons for A&E attendance, which is a limitation. The IMD score was included to indicate if this area based measure provided any additional predictive power above and beyond the individual measures. The findings that it is specifically poorer housing and unemployment that are the major factors that explain the IMD element is important for informing strategies to reduce A&E attendance in these areas. Information on GP consultations in the past 12 months was not included in the model as this could be a consequence and not necessarily a cause of A&E attendance. We also agree that it would be useful to link the present data to GP patient survey data to tease out potential reasons for the observed associations with healthcare access. This, however, was beyond the scope of the present study aim, which was to identify key and novel determinants. It should be noted, however, that appointment availability would not represent a confound of the distance variables we assessed, but rather could potentially represent an independent predictor of A&E attendance. This is also the case with other variables relevant to healthcare access, such as the quality of public transport links. We have suggested examining these constructs in future work on page 12.

VERSION 2 – REVIEW

REVIEWER	Jessica Sheringham UCL, United Kingdom
REVIEW RETURNED	25-Jun-2018

GENERAL COMMENTS	Our comments have been addressed to a large extent and the article is much clearer. There could still be greater clarity in wording. In particular in the abstract - would be helpful to describe more clearly analysis undertaken eg last line of methods - analysed rates of attendances? It is welcome to see strength of some associations in the results section of the abstract, but would be helpful to include figure for all of the relationships reported in the abstract. [a minor point but the selection of findings reported in the abstract is limited to those we might have expected but the paper finds some more surprising ones - eg age, lack of association with IMD]. I think there's a typo in the full paper results section in A&E distance sentence with the association given the wrong way round - or if it really does mean that living closer was associated with a lower rate of attendances, then this should be discussed differently.
--

REVIEWER	Sally Hull Queen Mary University of London UK
	I have written a recently published article on AED attendance
REVIEW RETURNED	24-Jun-2018

GENERAL COMMENTS	BMJ Open review 20 06 2018 What are the social predictors of Accident and Emergency attendance in disadvantaged d neighbourhoods? Results from a cross sectional household health survey in the North West of England. General comments: Thank you for the opportunity to review this paper again. The authors have made considerable improvements to the paper, in particular providing a lot more information on the questions asked and validated instruments used in the survey and providing baseline descriptors of the population. However the provision of this information does raise some further questions about the method used and how the interpretation of the results is framed. 1. The use of IMD. The advantage of your study is that you have detailed individual measures of housing, employment and some of the other measures which are captured at small area level within the IMD score. However the value of including the IMD score as an independent predictor is much less if all the areas studied were in the lowest 15% of IMD . The other question that arises is that of co-linearity between the IMD score and the outputs of the financial/housing/employment variables, did you do any analyses to explore this? You may be
--

	using more than one variable to measure the same underlying phenomena. I note that you classified employment in a binary way, although 25% of your sample were over 65 and of retirement age – did you do any sensitivity analyses to explore the effect of different categorisations? 2. Measurement of clinical long-term conditions and health status. The same questions relate to these clusters of variables. It seems highly likely that there may be co-linearity between the cluster of conditions you have chosen and the measures of the EQ-5D/mental health which you put in the model. As I understand it there may be a place for not putting all co-linear variables in the model because of the difficulty of interpretation. For example you include “any mental health issues” as a ‘condition’ and also have highly specific measures of depression/anxiety and paranoia. Having had the chance to look at the variables you have chosen it would appear that you have robust individual measures of mental health, of mobility and other factors in the EQ5D, however your health conditions are less well defined and not consistent with QOF nor linked to the GP record for accuracy. On P 12 you compare your results with that of Hull (12) suggesting that multimorbidity is not a risk for AED. There is strong evidence from multiple large studies over a long period of time that the burden of LTC is a predictor of AED use, whether measured for individuals (Hull) or using the aggregate measures of burden on practice/local area populations. I would suggest that the conclusions you draw from your results are rather too strong (P12). It is more likely that you have a relatively small sample, and that there is an element of co-linearity between your measures of health status, and that your measures of multimorbidity were less robust than the PHQ/EQ-5D. It may be more helpful to point out that these variables may actually be measuring more or less the same underlying factors which contribute to illness and frailty. 3. Measuring distance to the AED and the GP surgery. Distance from the AED is an important predictor of use. It was interesting to see the crude measure of distance from the GP surgery as an independent predictor in the adjusted analysis. However – depending on the geography, it may be that the hospital is a lot closer than the GP surgery, so that these variables are not independent. In the Hull study we include a map of hospital and GP practice locations (in the appendix) which illustrates this complexity. Simply moving GP surgeries to centres of population may not cut down attendance unless the AED is also moved further away! Further points. In the table you need to add adjusted analysis to the heading of column 1. In the discussion you have a lengthy (somewhat speculative) section on the education findings, but ignore the significant finding that the BME group consult at AED less. Similarly the negative findings around social support may be worth commenting on. The text needs proof reading as there are a number of You might consider adding to the Strengths and limitations a) possible recall bias of AED attendancesb) non-random sample
--	---

VERSION 2 – AUTHOR RESPONSE

Reviewer 1: a) There could still be greater clarity in wording. In particular in the abstract - would be helpful to describe more clearly analysis undertaken eg last line of methods - analysed rates of attendances? b) It is welcome to see strength of some associations in the results section of the abstract, but would be helpful to include figure for all of the relationships reported in the abstract. c) A minor point but the selection of findings reported in the abstract is limited to those we might have expected but the paper finds some more surprising ones - eg age, lack of association with IMD.	a) We have adjusted the wording throughout the manuscript to ensure our results are described with more precise language (i.e., predictors of number of A&E attendances or absolute risk ratios). b) All of the results reported in the abstract can be found in Table 1. c) We decided to focus on results related to employment, housing and healthcare access in the abstract as we believe these will be most useful and relevant for policy-makers.
I think there's a typo in the full paper results section in A&E distance sentence with the association given the wrong way round - or if it really does mean that living closer was associated with a lower rate of attendances, then this should be discussed differently.	We have left the results section as written because we believe it is reported correctly. Specifically, we state that "living further from an A&E department reduced the likelihood of attending an A&E service by seven percent per kilometre (RR = 0.93, 95%CI 0.89,0.97), and living further from a GP increased the likelihood of attending A&E by 46 percent per kilometre, after controlling for health status, socioeconomic and demographic factors (RR = 1.46, 95%CI 1.12, 1.90).
Reviewer 3:	
1. The use of IMD. a) The advantage of your study is that you have detailed individual measures of housing, employment and some of the other measures which are captured at small area level within the IMD score. However the value of including the IMD score as an independent predictor is much less if all the areas studied were in the lowest 15% of IMD . The other question that arises is that of co-linearity between the IMD score and the outputs of the financial/housing/employment variables, did you do any analyses to explore this? You may be using more than one variable to measure the same underlying phenomena. b) I note that you classified employment in a binary way, although 25% of your sample were over 65 and of retirement age – did you do any sensitivity analyses to explore the effect of different categorisations?	a) As noted in our previous response, there was adequate variability in the IMD variable to warrant inclusion in the model. Moreover, in addition to the variance estimates, it should be noted that the IMD index produced by the exponential transformation of each domain means that the most deprived 15% of neighbourhoods represent over 50% of the distribution of scores. We agree, however, with the second point made by Reviewer 3 that collinearity could be problematic for this variable due to conceptual and statistical overlap with individual-level measures of socioeconomic status (SES). To test whether this was the case, we regressed IMD score onto each of the individual-level SES variables in separate models, using dummy coding for categorical predictors. The largest standardised coefficient between IMD and any of the variables was .26 (for "debt struggle"). Thus, IMD could not be excluded from the model due to multicollinearity. This analysis is included as a footnote on page 11. b) We did not conduct any sensitivity analysis by re-categorising the employment variable. However, the fact that non-employment predicted A&E attendance while controlling for

	age suggests that the effect is not explained by age-related factors such as retirement.
2. a) Measurement of clinical long-term conditions and health status. The same questions relate to these clusters of variables. It seems highly likely that there may be co-linearity between the cluster of conditions you have chosen and the measures of the EQ-5D/mental health which you put in the model. As I understand it there may be a place for not putting all co-linear variables in the model because of the difficulty of interpretation. For example you include “any mental health issues” as a ‘condition’ and also have highly specific measures of depression/anxiety and paranoia. b) Having had the chance to look at the variables you have chosen it would appear that you have robust individual measures of mental health, of mobility and other factors in the EQ5D, however your health conditions are less well defined and not consistent with QOF nor linked to the GP record for accuracy. c) On P 12 you compare your results with that of Hull (12) suggesting that multimorbidity is not a risk for AED. There is strong evidence from multiple large studies over a long period of time that the burden of LTC is a predictor of AED use, whether measured for individuals (Hull) or using the aggregate measures of burden on practice/local area populations. I would suggest that the conclusions you draw from your results are rather too strong (P12). It is more likely that you have a relatively small sample, and that there is an element of co-linearity between your measures of health status, and that your measures of multimorbidity were less robust than the PHQ/EQ-5D. It may be more helpful to point out that these variables may actually be measuring more or less the same underlying factors which contribute to illness and frailty.	a) We included two different measures of physical health status in the model (EQ-5D and number of conditions/multimorbidity). While it could be argued that there is some conceptual overlap between general physical health status and number of conditions, the fact that both measures independently predict A&E attendance is, we argue, strong evidence that they are conceptually different and not collinear. In relation to the example provided, we are not aware of any reference to an “any mental health issues” variable in the manuscript. We only included specific validated measures of depression, anxiety, and paranoia. b) The health conditions we defined are consistent with those recognised in the Adult Psychiatric Morbidity Survey as noted in Appendix A. Because of the number of conditions examined, it was not possible to include all health conditions and hence the variable was re-categorised into number of conditions. Such methodological compromises were necessary for some measures, and we now acknowledge this as a limitation on page 13, stating: “Examining such a wide range of determinants meant striking a balance between including multiple determinants and examining those determinants in detail. Because of this, addressing more nuanced research questions, such as the role of different types of health conditions on A&E attendance, was not possible.” c) We have adjusted our language on page 12 to be more tentative regarding potential explanations for the divergent findings between our study and Hull et al.’s study. Specifically, we now state “One plausible explanation for this effect is that the number of health conditions a person has is not in itself a risk for A&E attendance – but this is just a marker of people with poorer mental health and health related quality of life. Further research is required to confirm whether this explains the divergent findings.” While it is plausible that some of our measures were more robust than others, we think it is unlikely that this explains our non-significant effect for multimorbidity because the examples provided (PHQ/EQ-5D/multimorbidity) measure the conceptually different constructs of depression, health-related problems, and number of conditions, respectively. The question of whether these may all tap an underlying latent construct is beyond the scope of this

	study and would require further analysis. We have not included a discussion of these issues as we have deemed other points of discussion to be of greater importance.
3. a) Measuring distance to the AED and the GP surgery. Distance from the AED is an important predictor of use. It was interesting to see the crude measure of distance from the GP surgery as an independent predictor in the adjusted analysis. However – depending on the geography, it may be that the hospital is a lot closer than the GP surgery, so that these variables are not independent. In the Hull study we include a map of hospital and GP practice locations (in the appendix) which illustrates this complexity. Simply moving GP surgeries to centres of population may not cut down attendance unless the AED is also moved further away! Further points. b) In the table you need to add adjusted analysis to the heading of column 1. c) In the discussion you have a lengthy (somewhat speculative) section on the education findings, but ignore the significant finding that the BME group consult at AED less. Similarly the negative findings around social support may be worth commenting on. d) The text needs proof reading as there are a number of typos.	a) To calculate distance variables, we aggregated distances at the LSOA level based on network road distances of each postcode to the nearest service. This method is the most accurate way of calculating distance to services that we are aware of. For example, it provides more accurate estimates when populations are condensed near the edges of LSOAs compared to methods used in other studies (e.g., Hull et al.). b) We have now corrected the column heading to read “Adjusted Rate Ratios”. c) Ethnicity is actually not significantly associated with A&E attendance in our study. Table 1 highlights that this is not significant (adjusted rate ratio of 0.60, non-s), only the absolute risk difference is significant, so that we did not focus on discussing this in the Discussion separately. We have now mentioned the negative finding on social support in the Discussion section on page 11. d) We have checked the manuscript again for typos and removed these. Thank you for highlighting this.
You might consider adding to the Strengths and limitations a) possible recall bias of AED attendences b) non-random sample	

VERSION 3 – REVIEW

REVIEWER	Dr Sally Hull Queen Mary University of London, UK Lead author for a study described in this report
REVIEW RETURNED	08-Sep-2018

GENERAL COMMENTS	What are the social predictors of Accident and Emergency attendance in disadvantaged d neighbourhoods? Results from a cross sectional household health survey in the North West of England. General comments: Thank you for the opportunity to review this paper again. The authors have made further improvements to the paper, in particular being more cautious about the interpretation of results from this survey.
--

	Ultimately the validity of this study does depend on the accurate measurement of the outcome measure (AED attendances in previous 12 months) and the predictor variables. I still have remaining concerns about measuring physical multimorbidity with the psychiatric morbidity survey which includes “any mental health issue” as no.6 in your appendix table of measures. The mental health domains are well measured using the well validated measures of depression, anxiety and paranoia. Other than this the authors have given helpful answers to the queries I raised previously. Further points. It is mould not mold on P5 You might consider adding to the Strengths and limitations a) possible recall bias of AED attendances b) non-random sample
--	---

VERSION 3 – AUTHOR RESPONSE

Thank you very much for the opportunity to address the minor additional changes suggested by the reviewer.

The reviewer's first point about the measurement of physical multimorbidity is interesting, and as always there can be a trade off between having a large survey with all measures that would be preferred to be included and the return rate, with lower return rates the longer the survey. In the present analysis, we have not used physical multimorbidity as a variable, but have provided this information on the measure in the Appendix, as pointed out by the reviewer. We have therefore not made any changes to the article but consider this an interesting point.

We have changed 'mold' into 'mould'.

We have now further clarified the potential recall bias of A&E attendance in the Limitations.

The reviewer suggested to discuss the limitation of the non-random sampling. However, as stated on page 5, we have used a random sampling technique for both the neighbourhoods as well as the individual households.